# Selenium in Peptide Chemistry

**DOI:** 10.3390/molecules28073198

**Published:** 2023-04-04

**Authors:** Özge Pehlivan, Mateusz Waliczek, Monika Kijewska, Piotr Stefanowicz

**Affiliations:** Faculty of Chemistry, University of Wrocław, F. Joliot-Curie 14, 50-383 Wrocław, Poland

**Keywords:** selenium, native chemical ligation, stapled peptides, photochemical reactions

## Abstract

In recent years, researchers have been exploring the potential of incorporating selenium into peptides, as this element possesses unique properties that can enhance the reactivity of these compounds. Selenium is a non-metallic element that has a similar electronic configuration to sulfur. However, due to its larger atomic size and lower electronegativity, it is more nucleophilic than sulfur. This property makes selenium more reactive toward electrophiles. One of the most significant differences between selenium and sulfur is the dissociation of the Se-H bond. The Se-H bond is more easily dissociated than the S-H bond, leading to higher acidity of selenocysteine (Sec) compared to cysteine (Cys). This difference in acidity can be exploited to selectively modify the reactivity of peptides containing Sec. Furthermore, Se-H bonds in selenium-containing peptides are more susceptible to oxidation than their sulfur analogs. This property can be used to selectively modify the peptides by introducing new functional groups, such as disulfide bonds, which are important for protein folding and stability. These unique properties of selenium-containing peptides have found numerous applications in the field of chemical biology. For instance, selenium-containing peptides have been used in native chemical ligation (NCL). In addition, the reactivity of Sec can be harnessed to create cyclic and stapled peptides. Other chemical modifications, such as oxidation, reduction, and photochemical reactions, have also been applied to selenium-containing peptides to create novel molecules with unique biological properties.

## 1. Introduction

Se-containing proteins (SePs—selenoproteins) were identified in mammals and higher plants and play important roles in biological processes [1,2,3,4,5]. In mammals, most of the identified SePs are enzymes involved in the regulation of lipid membrane oxidation and thyroid hormones, the NADPH-dependent reduction of thioredoxin, and muscle metabolism [2,3]. The biological study revealed that selenoproteins, mainly containing Sec, are characterized by higher enzymatic efficiency and redox potentials than similar proteins containing Cys, resulting in stronger resistance to oxidation in humans [6]. Moreover, studies regarding the biological significance of selenoprotein P showed the correlation of the presence of this protein with type 2 diabetes and cardiovascular diseases [7,8]. Generally, SePs are involved in numerous redox processes due to their unique abilities to react with oxygen and related ROS in a readily reversible manner [9]. Therefore, selenium-containing proteins have become an object of interest in the field of chemical biology. Sec is a genetically encoded amino acid in all domains of life, but not in all organisms. Higher plants, for example, are a large group that do not have Sec genetically encoded, and Sec is incorporated into proteins non-specifically through the metabolic pathway of sulfur analog, resulting in the replacement of Cys [10]. The translational Sec incorporation is based on the alternative use of the stop codon UGA and requires the presence of an mRNA structure called selenocysteine insertion sequence (SECIS) element [11]. This motif directs the cell to translate UGA codons as selenocysteines. In bacteria, SECIS is located in the coding region immediately after the Sec-encoding UGA codon, while in eukaryotes, it occurs in the three prime untranslated region (3′-UTR) of an mRNA. The complicated biosynthesis of SePs and low yield natural expression makes access to these molecules limited. This situation has contributed to the rapid development of various techniques for expressing proteins containing selenocysteine, including mammalian, bacterial, and cell-free systems. A recent paper reviews these techniques [12]. The attention attracted by selenoproteins has also led to the rapid development of methods for the synthesis of selenopeptides, including solid-phase peptide synthesis (SPPS) and native chemical ligation. Currently, methods allowing the synthesis of these compounds are well developed and have been the subject of numerous review papers [13]. The peptides containing selenium were found in a wide range of applications. Selenopeptides are substrates that can be used for protein synthesis by NCL. The incorporation of Sec allows selective post-synthetic modification of the peptide. Sec-containing peptides are used to study folding pathways, induce selective folding, and as probes in NMR spectroscopy and radio labeling.

Sec-containing peptides have been shown to exhibit antimicrobial, immunomodulatory, and anticancer activities and can be used as models for active centers of enzymes. For example, the selenium-containing pentapeptide Sec-Arg-Gly-Asp-Cys showed glutathione peroxidase (GPx) activity. Interestingly, the conjugation of this peptide to gold nanoparticles results in a 14-fold increase in GPx activity compared to the free peptide. In addition, the conjugate obtained exhibits enzyme-like kinetics. Therefore, gold nanoparticles modified with selenopeptide can be considered as an enzyme mimetic [14]. The conjugates of selenopeptides with nicotinic acid induce the formation of mesotubes—structures with potential applications in nanotechnology and medicine [15]. The molecular self-assembly of these compounds requires the presence of both a nicotinic acid moiety and a selenium atom. The unique properties of Sec-containing peptides make them an attractive target for the development of new therapeutic agents with improved efficacy and specificity. A recent paper gives an overview of the biological applications of selenopeptides [16].

Overall, selenopeptides offer a wide range of opportunities for the development of new chemical and biological tools and advance our understanding of biological processes and the development of new therapeutic agents. This review covers the specific chemical properties of peptides containing selenium and their potential for numerous selective chemical reactions, including selenium-based NCL, selective formation of diselenide bridges in peptides, application of Sec for peptide stapling and cyclization, and photochemical reactions of selenopeptides.

## 2. Selenium and Sulfur Comparison of Properties and Reactivity

Selenium and sulfur are both chalcogen group elements that share similar chemical properties in terms of their ionic radii and electronegativity [17]. However, significant differences also differentiate those elements in their chemical reactivity. Pleasants et al. [18] demonstrated that the exchange reaction between cysteamine and cystamine at physiological pH was 1.2 × 10^7^ times slower than that of selenium-containing analogs, attributing to the better nucleophilicity and leaving group ability of selenolate (RSe^−^) compared to thiolate (RS^−^), which resulted in faster nucleophilic reactions. Similarly, Steinmann et al. [19] studied the nucleophilic and electrophilic properties of selenium and sulfur in a thiol/disulfide-like exchange reaction, where selenium acted as both an electrophile and a nucleophile much faster than the sulfur analog, with differences of 4 and 2–3 orders of magnitude, respectively. Furthermore, selenocystine-containing cyclic peptides were reported to exhibit higher reactivity toward glutathione_red_ (GSH) than the corresponding disulfides due to several factors, including lower energy of the diselenide bond, better-leaving group ability, and greater nucleophilicity of the selenolate at pH 7.5 [20]. Thus, the better nucleophilicity of selenolate in comparison to its sulfur analog accounted for the readily disassociated form of the selenol group at physiological pH, while the thiol group is mostly in the protonated form, which indicates the higher acidity of the SeH group with respect to the SH group. The reported pKa values of selenocysteamine (5.0) [21] and Sec (5.24) [22] are approximately three units lower than those of cysteamine (8.37) [23] and Cys (8.22) [24], respectively.

The redox potential is another significant difference between selenium and sulfur that was investigated by Besse et al. [25], who conducted a study on a selenium-containing analog of the Grx-octapeptide [Cys11,Cys14]. Their results revealed the high stability of the diselenide bond toward reduction with dithiothreitol_red_ (DTT) at pH 7 compared to the disulfide analog, regarding the relative concentration of reduced and oxidized species measured by circular dichroism (CD) analysis. The redox potential of the diselenide bond (E_o_ = −381 mV) was determined to be lower than that of the disulfide bond (E_o_ = −180 mV) based on the calculations obtained from the K_ox_ reference to glutathione, a redox reagent widely used to facilitate protein folding [26,27]. Beld et al. [28] demonstrated that the diselenide bond in a selenium-containing analog of glutathione (GSeSeG) is highly stable, which is associated with its lower redox potential (E_o_ = −407 mV).

Se-Se bonds possess a bond disassociation energy (172 kJ/mol^−1^) that is lower than S-S bonds (240 kJ/mol^−1^) [29]. In addition, selanyl radicals are much less reactive than thiyl radicals. While thiyl radicals react with tyrosine and tryptophan moieties, selanyl radicals are inefficient in carrying out this reaction [30]. Similarly, Cα-H abstraction, a reaction typical for thiyl radicals formed in Cys-containing peptides, is extremely slow in their selenium analogs.

Consequently, despite their similarities, selenium and sulfur exhibit distinct chemical reactivity in certain contexts, particularly with respect to their nucleophilicity, electrophilicity, acidity, and redox potentials. In addition, the lower bond dissociation energy of the Se-Se compared to the S-S makes diselenides more prone to homolytic cleavage and subsequent generation of radical species [31]. These differences in chemical reactivity may, therefore, have implications for various biological and chemical processes, where selenium and sulfur are involved, such as in redox reactions or in the formation of chemical bonds with other elements.

## 3. Selenium-Mediated Native Chemical Ligation

There are many different methods of peptide assembly that have been developed over the years, ranging from traditional SPPS to more novel techniques, such as fragment condensation and NCL. SPPS is one of the most widely used methods for peptide assembly. Despite the huge number of described procedures, methods of peptide synthesis, and new resins developed, one fact remains unchanged: as the length of the peptide chain increases, the yield of the obtained product decreases. For this reason, the solid-supported peptide synthesis for sequences longer than 40–50 amino acid residues is considered inefficient [32]. The iterative amino acid couplings that do not proceed with 100% yield, particularly for sterically hindered amino acids, result in the accumulation of many difficult-to-purify by-products. Thus, the overall yield of the target peptide is unsatisfactory. The development of NCL in the 1990s was a significant milestone in the field of peptide synthesis [33]. NCL is a powerful synthetic strategy that allows for the efficient and selective ligation of two unprotected peptide fragments under mild conditions, without the need for specialized resins or protecting groups, allowing access to long polypeptide chains and small proteins [34]. This two-step approach involves the synthesis of SPPS of two short peptide fragments—one containing a C-terminal thioester and the other containing an N-terminal Cys residue (Figure 1). The products cleaved from the solid support are then subjected to NCL between the thioester and the alpha-amino group of a Cys residue.

This conjugation proceeds in an aqueous environment in the presence of a thiol catalyst, e.g., 4-mercaptophenylacetic acid (MPAA) or 2-mercaptoethane sulfonate sodium (MESNa). Importantly, this reaction proceeds chemoselectively and requires no side-chain protection. The developed chemical tool enables access to the so-called total protein synthesis [35]. It is obvious that biotechnological methods routinely enable the production of various proteins with satisfactory yields, while total chemical synthesis enables the incorporation of additional functionalities that are impossible to obtain by conventional bioengineering. Examples include custom post-translational modifications, non-proteinogenic amino acids, or fluorescent tags. Since the original report, NCL has been extensively explored in peptide chemistry and has been employed in the synthesis of hundreds of proteins [36,37]. Due to the high utility of this methodology, the initially presented NCL has undergone many modifications including reaction conditions, e.g., thiol catalysts, or replacement of the C-terminal thioester with a thioester surrogate [38]. With the advances of NCL, attention has been paid to addressing some inherent limitations of this methodology. One of the drawbacks concerns the lack of chemoselectivity during the desulfurization reaction in the presence of native Cys residues. Importantly, the average reaction rates are relatively long, especially at sterically hindered amino acid junctions [39]. To overcome these limitations, the application of Sec in NCL-like transformations with peptide thioester was independently demonstrated by three research groups [40,41,42]. As a consequence of the lower oxidation potential of Sec than Cys, it occurs exclusively as a diselenide dimer [43,44]. Therefore, the presence of a reducing agent (e.g., DTT, TCEP) during ligation is inherent, as only monomeric selenolate takes part in this reaction. Better nucleophilicity of Sec in comparison to its sulfur analog significantly affects the ligation rate and leads to a significant reduction in reaction time. The lower pKa of Sec is another advantage in performing the reaction at a lower pH, minimizing the risk of selenoester hydrolysis, which frequently occurs during conventional thioester-based NCL, resulting in a decrease in ligation yield. Recently, Sayers et al. [45] reported a Sec-based NCL-employed assembly of peptide nucleic acids (PNA). The authors used the C-terminal selenoester and N-terminal Sec (reduced in situ during reaction conditions) in NCL. Additionally, the conjunction formation was supported by the template effect resulting from the recognition of complementary nucleobases. Thus, according to this study, a combination of selenium-based NCL with templation impressively increased the ligation rate—10 times faster than traditional NCL at Cys. Moreover, this technology was then employed in a paper-based lateral flow assay for the rapid and sequence-specific detection of oligonucleotides, including miRNA, in cell lysates. A few years ago, Chisholm et al. [46] showed that the reductive diselenide-selenoester ligation (rDSL) method enables efficient ligation of peptide fragments down to low nanomolar concentrations. This finding was demonstrated with a highly efficient photodeselenization process, which affords native polypeptides. Interestingly, Mitchell et al. [47] reported selenoester-selenocystine peptide ligation that proceeds rapidly without any additive, such as 4-selenophenylacetic acid (selenium analog of MPAA) (Figure 2). Although the authors proposed a redox associative mechanism, the exact course of the reaction has still been under investigation. However, recent studies [46] provide a simple and convincing mechanism for diselenide-selenoester ligation (DSL). The proposed pathway involves the formation of a small amount of phenylselenoate from the peptide selenoester. This compound participates in a redox exchange with a peptide diselenide dimer, forming a reduced selenopeptide. The reaction product then reacts with a peptide selenoester via an NCL-like mechanism (transselenoesterification followed by a Se-to-N acyl shift), which generates a native amide bond and propagates the DSL reaction. The proposed mechanism is supported by the catalytic effect of benzeneselenol on selenoester-diselenide ligation. A high reaction rate was achieved using phenylselenoesters, even at bulky junctions.

While presenting the above content on peptide selenoesters, it is also necessary to mention the methods of synthesis of these compounds, which, due to their less common use in NCL methodology, have been developed to a much lesser extent in comparison to peptide thioesters [48]. Jakubke et al. [49] showed for the first time an efficient protocol for the solution-phase synthesis of side-chain protected peptides containing phenylselenoester in which a mixed anhydride or carbodiimide activation was used. An interesting development, however, was similarly the usage of side-chain protected peptide followed by treatment with diphenyldiselenide (DPDS)/triphenylphosphine. This method was then adopted on solid support by Hanna et al. [50]. Another example of solid-phase peptide selenoester synthesis with a specially designed linker was demonstrated by Ghassemian et al. [51].

The original NCL methodology has some drawbacks, as it is limited to Cys-containing peptides. It should be noted that Cys is the least abundant proteinogenic amino acid (1.8%), thus the number of possible ligations is significantly limited. However, it is possible to perform desulfurization to obtain alanine, an amino acid that is very common in polypeptide sequences (8.9% abundance). Such an approach provides access to the assembly of most polypeptides or proteins, especially Cys-free ones [52]. Nevertheless, this methodology is also not without flaws. When a peptide sequence contains several cysteinyl residues, the aforementioned desulfurization may proceed in a non-selective manner. Therefore, the side chain protection of Cys residues not involved in ligation is required. Furthermore, standard desulfurization protocols involve the usage of a large excess of Raney nickel [53], palladium, or—giving better yields—TCEP with a radical initiator [54]. This issue, however, has been overcome by Gieselman et al. [40], who demonstrated an efficient deselenization method for Sec-based peptides. This elimination was driven by the presence of mild reducing agents, such as TCEP and DTT, as hydrogen donors [55]. Contrary to the conventional NCL, the deselenization has been characterized as highly chemoselective, which retains the remaining Cys residues pivotal for the structure of the target protein. Another interesting discovery presented independently by two research groups (Dery et al. [56]; Malins et al. [57]) concerns the possibility of converting Sec to native serine using TCEP and the exogenous oxidants (the difference between these studies concerns the oxidizing reagents used). It is worth noting that these findings broaden the Sec-based NCL, not only to alanine but also to serine junctions.

The use of selenium-mediated NCL has generated significant interest in the scientific community involved in protein and peptide synthesis. Due to the large number of applications, we will present only a few examples of Sec-mediated NCL in a polypeptide or protein assembly. Hondal et al. [42] demonstrated an early example of using Sec-mediated NCL to synthesize ribonuclease A in combination with protein expression. The authors used rDNA technology to produce a fragment corresponding to residues 1–109 with a C-terminal thioester. The resulting fragment was ligated to a synthetic peptide containing a Cys or Sec moiety. More than a decade ago, the first articles describing the Sec-based NCL-deselenization methodology were published, which focused on relatively simple polypeptides, including a 38-residue fragment of the redox enzyme glutaredoxin 3 (Grx3, 1–38) [53]. A particularly interesting example, showing the extraordinary possibilities of this method, was the assembly of the challenging protein, 125-residue human phosphohistidine phosphatase 1 (PHPT1), which has three Cys residues near the C-terminus [58]. The authors of this study exploited consecutive Sec-based ligation and deselenization. For this purpose, three unprotected peptide segments were used, which were then combined with two NCL reactions at alanine and Cys junctions. The middle segment, however, was prepared as a selenazolidine being a masked precursor of the Sec. Prior to target ligation, selenazolidine was converted to reactive Sec by treatment with *O*-methylhydroxylamine. Similarly, the total chemical synthesis of two natural selenoproteins, selenoprotein M (SELENOM) and selenoprotein W (SELENOW), was executed [59].

Recent advances in the semi-synthesis of selenoproteins using protein expression, SPPS, and Sec-based native ligation have been reviewed in a recent paper reported by Chung et al. [12]. Another possibility offered by selenium chemistry is the use of peptides with a Sec moiety as surrogates for selenoesters. Melnyk’s group has shown that peptides containing Sec are susceptible to acyl transfer from the amide nitrogen to the selenium atom. This rearrangement leads to the formation of selenoesters, which undergo NCL with N-terminal Cys or Sec, resulting in transamidation or peptide metathesis [60].

## 4. Redox Properties of Selenopeptides—Selective Formation of Se-Se Bridges in the Presence of S-H Groups

Disulfide bonds play a significant role in the folding and stabilizing of protein structures by lowering the entropy of the denatured/unfolded state [61]. Accordingly, folding is dictated by the oxidation, reduction, and reshuffling of disulfide bonds via thiol/disulfide exchange reactions to adopt the native conformation of proteins [62]. On the other hand, the slow kinetics of disulfide bonds can impede the folding process and consequently reduce the yield of properly folded proteins when multiple rearrangements of the intermediates are entailed [63]. Since Sec can be readily oxidized to the corresponding diselenide form, unaffected by the presence of other protein thiols or reducing agents [64,65], in vitro protein folding has been extensively investigated to gain insight into how the substitution of Sec for Cys affects the folding and function of sulfur-containing proteins [66]. Walewska et al. [67] demonstrated that pairwise replacement of Cys with Sec in µ-conotoxin SIIIA enhanced the accumulation of natively folded proteins and decreased the number of intermediates. The substitution of the diselenide crosslinker for the interchain disulfide bridge (A6–A11) in human insulin was reported to ameliorate the challenge of efficiently combining the A and B chains to yield native insulin in a relatively short time and with a higher yield, which was accompanied by increased stability of the protein compared to the wild-type analog [68], as also shown in caenopore-5 (Cp-5) [69], human epidermal growth factor (EGF) [70], and α-conotoxin [71] studies. Interchain substitution on a single chain is indeed a more straightforward strategy than the synthetic method developed by Arai et al. [72], who modified both A and B chains ([C7U^A^,C7U^B^]) of bovine pancreatic insulin (BPIns).

Cys-to-Sec substitution in proteins is advantageous, as it maintains the function of the protein and its 3D structure in most cases [73,74]. However, it is important to consider that different analogs of the same protein may not interact equally effectively with the target receptor. Walewska et al. [75] presented a study in which the inhibitory activity of [U^2,19^] ([U^2,19^]) indicates the substitution in the parent protein (EETI—Ecballium elaterium trypsin inhibitor). The amino acids in positions 2 and 19 were replaced by unusual amino acid “U”—Sec. EETI-II toward bovine β-trypsin was 2-fold less efficient in comparison to [U^9,21^] EETI-II and [U^15,27^] EETI-II, although all seleno analogs retained the biological activity. The weak binding interaction of [U^2,19^] EETI-II was associated with the increased size of the diselenide bridge that was adjacent to the inhibitory binding loop, thereby anticipating the causing of conformational changes in the loop.

The size of diselenide crosslinkers is crucial not only for protein function but also for determining the efficacy of native folding. Gowd et al. [76] explicated the impact of the size of diselenide and implicitly non-native disulfide crosslinkers on the yield of native ω-conotoxin GVIA [C1U,C16U] (The “[C1U,C16U]” part indicates that two specific Cys residues within the peptide were replaced by unusual amino acids called “U” (which stands for “Sec” in this context)), [C8U,C19U], and [C15U,C26U]. Specifically, the larger inner size of the diselenide bridge was found to increase the yield of native folding, which was observed when the inner size of the non-native disulfide bridges in GVIA [C8U,C19U] was greater and similar, as the accumulation of non-native species was presumably precluded. On the other hand, the smaller inner size of the non-native disulfide bridges in GVIA [C1U,C16U] and GVIA [C15U,C26U] resulted in a decrease in the yield of native folding due to the accumulation of non-native species, which likely disrupted the native folding pathway.

Steiner et al. [77] studied the in vitro oxidative folding of µ-selenoconotoxin SIIIA in the absence of an oxidative reagent. Their findings showed an increase in the rate constants of initial thiol oxidation (k_ox_) and native state formation (k_native_) compared to the wild-type analog, which was also supported by the experiments conducted on ω-conotoxin GVIA [C8U,C19U], indicating that the intramolecular diselenide bridge was able to catalyze the oxidation of thiols in the presence of only molecular oxygen. Considering the oxidative role of selenocystine and CuCl_2_ in thiol oxidation, the folding kinetics of WT-GVIA and its selenium-containing analog, GVIA [C8U,C19U], were intermolecularly determined. Unfortunately, the oxidation of GVIA [C8U,C19U] in the presence of selenocystine and CuCl_2_ did not explicitly affect the k_ox_ and k_native_. On the other hand, the folding of WT-GVIA in the presence of selenocystine was less potent than the folding of GVIA [C8U,C19U] under air oxidation. Furthermore, the use of an oxidant in promoting the folding of wild-type Hirudin (WT-Hir) was significant, as only 27% of the protein reached its native state in the absence of GSSG, whereas the selenium-containing analog, Hir(C16U/C28U), folded with an 80% yield in the presence of only molecular oxygen, although the oxidant-free folding of Hir(C16U/C28U) was comparable to that of WT-Hir in the presence of GSSG. This study indicated that not only native diselenide crosslinkers but also the non-native substitution, Hir(C6U/C16U), accelerated the folding rate, although the early stage intermediate (1-SS) was not populated [78]. A similar approach was utilized by Metanis et al. [79], who altered the folding pathway of bovine pancreatic trypsin inhibitor (BPTI) via non-native diselenide substitution at position [5,6,7,8,9,10,11,12,13,14]. This induced the formation of a non-native intermediate, [5,6,7,8,9,10,11,12,13,14,30,31,32,33,34,35,36,37,38,39,40,41,42,43,44,45,46,47,48,49,50,51], which was less stable than the kinetically trapped intermediates, N’ and N*, of BPTI. Therefore, the selenoprotein reached the native state faster than its BPTI analog, resulting in undetectable N’ and N* species, presumably due to their rapid conversion to the native state. The authors also examined the effect of a single Sec substitution at position [5] on the foldability of BPTI. Their results showed that the folding of C5U BPTI was completed within 3 h, which was significantly faster than that of the wild-type analog that reached the native state over 21 h without affecting the folding pathway [80].

In addition to studies concerning Sec-assisted intramolecular protein folding, GSeSeG was found to be a more potent oxidant than GSSG in promoting RNase A folding under different redox conditions. Mainly, it required 10-fold less redox buffer to achieve comparable yields at pH 8.0. Although GSSG was unable to promote folding at pH 5 due to the protonated form of the Cys thiols, GSeSeG induced a native state with a 75% yield in 68 h [81], indicating that small diselenides are an effective means of catalyzing disulfide bond formation. Reddy et al. [82] synthesized diselenide-containing small molecules (Figure 3) to investigate their efficacy in enhancing the folding of BPTI. Their results showed the folding kinetics of BPTI were 10-fold better in the presence of diselenide 2 and 3 than GSSG. Among all diselenides, the folding rate of BPTI was slower in the presence of diselenide 1, presumably due to the steric hindrance of the molecule. Nonetheless, diselenide 1 was still able to enhance the overall folding of BPTI, despite having slower folding kinetics in the initial stages of BPTI folding compared to GSSG.

## 5. Identification and Detection of Selenopeptides

The identification of selenoproteins is quite challenging due to the low concentration of these proteins and the effect of complex matrix ions [83]. Therefore, preconcentration based on ultrasonic-assisted extraction followed by separation by chromatographic methods, such as reversed-phase (RP), IEC, and SEC, was used [84]. The selenium content in proteins was determined by high performance liquid chromatography (HPLC) combined with ICP-MS as a detector (HPLC-ICP-MS), while analysis by HPLC-ESI-MS/MS as a conventional proteomics method, preceded by enzymatic hydrolysis of proteins, provides molecular mass, structural information, and modification sites based on the unique isotope distribution for selenopeptides (SePPs) [85,86,87]. Another approach to the detection of Sec-containing peptides is based on the differences in nucleophilicity and acidity between SeH and SH groups. At pH 4, Sec moieties react selectively with iodoacetyl-PEG2-biotin, while Cys residues remain unreacted. The biotinylated peptides were separated on avidin and detected by LC-MS [88].

Alternatively, another method has been proposed for the detection of selenoproteins. After derivatization of Sec moieties in proteins with an iodoacetamide-alkyne probe at low pH, proteins with Sec are attached to a diazo-biotin-azide linker by copper-catalyzed azide-alkyne cycloaddition and then enriched on streptavidin beads. Proteins are subjected to on-bead trypsin digestion and selectively eluted from the beads for LC/LC-MS/MS identification [89]. Although not fully selective, this procedure provides high enrichment for Sec.

The identification of peptides from different species using different coupling techniques confirmed that the predominant metabolites of Se in SePs were selenomethionine (SeMet), Sec, and Se-methylselenocysteine (MeSec) [84].

## 6. Selenopeptides Cyclization and Stapling

The identification of new modification sites in proteins led to the development of methods for selenopeptides synthesis, opening new possibilities for studying biological activity [72]. The development of the synthesis on the solid support according to the Fmoc strategy and the development of the synthesis methods of appropriate amino acid surrogates containing selenium instead of sulfur in Cys and appropriate orthogonal protecting groups enabled the synthesis of any selenopeptide sequences [50,73,90]. Peptides containing one Sec in the peptide sequence are immediately oxidized after removal from the solid support, resulting in the linear diselenide form. Utilizing the Sec properties, several reports regarding the new synthetic methods of cyclic and stapled peptides have been developed [90,91,92]. The precisely designed architecture of such peptides has an impact on the properties causing conformational rigidity correlated with increasing biological activity and stability against proteolytic degradation [93]. De Araujo and co-workers [91] proposed the cyclization of peptides by using selenolanthionine bridges. This method was based on the Se-alkylation reaction between the Sec substituted for Cys in oxytocin and β-chloroalanine introduced in one peptide sequence (Figure 4). Moreover, this strategy was successfully applied to the synthesis of an overlapping double selenoether cyclized peptide (α-conotoxin ImI), where both native disulfide bonds were replaced by selenolanthionine bridges. The lanthionine linkage is characterized by greater stability in redox processes, which may have a critical impact on bioactivity, and lead to a significant decrease in its agonist activity [94].

The higher side-chain acidity of Sec vs. Cys was used in reactions with electrophilic alkanes, resulting in crosslinking within unprotected linear peptides under mild aqueous conditions. De Araujo and co-workers [92] proposed a two-component selenoether stapling method based on crosslinking of two Sec incorporated at positions (i, i + 4), (i, i + 7), or (i, i + 11) in an unstructured peptide analog of the tumor suppressor by alkylating agents with different length, reactivity, hydrophobicity, and rigidity to induce helicity in a bioactive peptide (Figure 5). In each case, they noticed a significantly higher helicity for stapled analogs than for unconstrained peptides. For the (i, i + 4) series, the most helical stabilization was observed for the o-xylene linkage containing an 8-atom-length staple and then for aliphatic alkylating agents having the same length. In the case of the (i, i + 7) and (i, i + 11) series, the strongest helical stabilization occurred for aliphatic crosslinkers containing an 8–9-carbon-length and a 10-carbon-length, respectively. Moreover, the analogs from the (i, i + 7) series showed the most active and reduced cell viability of MCF-7 breast cancer cells compared to control linear peptides.

## 7. Modified Peptides Containing Selenium

The Ugi reaction is a versatile approach employed for the derivatization of peptoids and peptide arrays [95,96], the design of peptide–peptoid fusion [97], and the synthesis of amino acid-based polypeptoids [98], inducing the formation of Ugi products/peptide-like molecules by the reaction of an aldehyde or a ketone, a carboxylic acid, and an amine in a one-pot process [99]. Abbas et al. [100] reported the synthesis of Sec-containing peptoids via Ugi four-component reaction (Ugi-4CR). To construct a Sec moiety, the seleno group was appended to one of the four components, which bears a carbonyl functional group. The synthesis of the component was carried out sequentially by the reaction of KSeCN with 1,1-diethoxy-2-bromoacetate and reductive alkylation, affording the formation of selenoacetals, whose acidic treatment yielded selenoaldehydes. The experimental results revealed that the yield of Ugi reactions substantially depended on the reaction conditions and the choice of amines used. Reactions performed under microwave heating were superior in organic solvents to those carried out at room temperature. On the other hand, an aqueous reaction medium at room temperature prompted the highest yield formation of products, which also provided an advantage to using selenoacetals in the protected form.

Aziridines are three-membered heterocycles found in the structure of natural compounds [101] and harnessed to design biologically active synthetic molecules [102,103]. Due to their capability of undergoing nucleophilic ring opening, aziridines serve as intermediates in the synthesis of amino acid derivatives and peptides using nucleophiles such as thiols [104,105], amines [106], and selenols [107,108] in the ring-opening process. Braga et al. [107] described the utility of aziridine precursors for the preparation of selenium-containing amino acids. The synthesis was achieved by converting α-amino alcohols to N-Boc aziridines that underwent nucleophilic ring opening at the less hindered carbon, yielding chiral β-selenoamine moieties. The nucleophile was generated by the reaction of diphenyl diselenide with NaBH_4_ to give a corresponding phenyl selenide anion. The resulting selenoamines were successfully assembled with Boc-protected L-valine, L-phenylalanine, and L-proline by building small peptide libraries in high yields without epimerization at the chiral centers.

Arsenyan et al. [109] proposed a facile method to derivatize Sec-containing peptides via the 5-endo-dig and 6-endo-dig cyclization reactions that were discovered in 1976, and whose prefixes denote the number of atoms in the forming ring (5 and 6), the relative position of the bond being broken to the newly formed ring (endo), and the hybridization at the ring closure site (dig) [110,111]. According to the postulated mechanism, the selenium electrophile was generated by the coordination of Sec-containing peptides with a Lewis acid, copper (II) bromide (CuBr_2_), which was subsequently reacted with 2-propargyl *N*-pyridines to form indolizinium derivatives and with 2-ethynylbiaryls to form their polyaromatic systems via 5-endo-dig and 6-endo-dig cyclization, respectively. The method was further extended, involving an oxidant, K_2_S_2_O_8_, to generate a selenium electrophile for the synthesis of benzo[b]furans and indoles derivatives via 5-endo-dig cyclization [112], as well as coumarin and quinolinone derivatives via 6-endo-dig cyclization [113].

## 8. Derivatization Based on Oxidation/Elimination: Conversion of Selenocysteine to Dehydroalanine

Dehydroalanine (Dha) is a non-proteinogenic amino acid that constitutes the structure of naturally occurring peptides, including nisin [114], siomycin [115], and thiostrepton [116]. An early experimental study described the formation of Dha by oxidative elimination of selenium from Sec derivatives [117]. Later on, Sec-containing peptides were chemoselectively converted to Dha precursors [118], enabling the synthesis of biologically active compounds that possess a Dha skeleton, such as alternariolide (AM-toxin I) [119]. Dha formation induced by Sec has also been demonstrated in in vitro [120] and in vivo studies [121]. The resulting Dha moiety is a versatile tool that is susceptible to further chemical modification [122] and is used to engineer the properties of proteins for various purposes, including protein–protein crosslinking [123], protein labeling [124], and antibacterial activity [125].

## 9. Photochemical Reactions of Selenium-Containing Peptides

Due to the lower bond disassociation energy of Se-Se, homolytic scission and/or association of the bonds within and/or between the molecules occur under milder conditions than those of S-S analogs [126]. While visible light can promote the formation of asymmetrical diselenides, interchanging the partners of symmetrical ones, such as diselenide-containing small molecules [126,127], polymers [128], and peptides [129], this stimulus is insufficient to rupture S-S bonds, hence requiring bond activation by a strong trigger, namely UV light [31,130]. Waliczek et al. [129] demonstrated a visible-light-induced exchange reaction of Sec-containing homodimeric peptides. Experiments conducted using an equimolar concentration of the sample containing either a pair of linear homodimers (Figure 6) or peptide libraries showed the formation of heterodimers with high efficiency within 30 min. Prolongation of the irradiation time to 24 h resulted in a significant increase in the yield of heterodimer formation.

The authors further reported the UV light-mediated formation of selenolanthionine linkage from selenocystine-containing cyclic peptides (e.g., nisin analog), which resulted in the elimination of one selenium atom from the diselenide bridge (Figure 7). LC-MS analysis revealed 76% conversion of diselenide to selenoether bond within 1 h of irradiation. Moreover, the formation of a selenolanthionine bridge from cyclic peptides involving both intramolecular diselenide and disulfide bonds was achieved in 10 min of irradiation without affecting the disulfide bond [90]. A similar reaction was later described by Dowman et al. [131], who demonstrated the photocatalytic diselenide contraction to yield selenoethers. This transformation was induced by irradiation of Sec-containing dimers in the presence of the phosphine 1,3,5-triaza-7-phosphaadamantane (PTA) and the iridium photocatalyst, [Ir(dF(CF_3_)ppy)_2_(dtbpy)]PF_6_, under LED_450_ irradiation within 5 min.

The straightforward visible light-initiated reaction leading to Sec-containing indole-based macrocycles via intramolecular Se-C bond formation was proposed by Lapcinska et al. [132]. In the presented study, Boc-Sec-containing dipeptide or tripeptide dimers attached to the C4 or C5 position of indole through an ester or amide bond were irradiated with LED_460_ light in the presence of transition metal-free photocatalyst RB, Rose Bengal, leading to macrocyclization (Figure 8A). A second method did not directly concern the cyclization through new Se-C bonds between the Sec residue and the indole located in one peptide chain, but indirectly introduced the appropriately substituted indole by selenylation in the same conditions as previously and subsequently intramolecular amide bond formation (Figure 8B). A detailed mechanistic study showed that the selenium radical formed under light irradiation was converted to a selenium electrophile, which reacted with electron-rich N-heterocycles, resulting in Sec-containing indoles.

The authors later described the synthesis of selenosulfide bond-containing peptides driven by visible light. The reaction of Sec-containing dimers with glutathione (GSH) in the presence of Rose Bengal induced the formation of selenosulfide bonds through the sulfur-centered radical, which was generated under LED_460_ (Figure 9). It is worth noting that visible light was essential for the reaction, as it did not proceed in either daylight or darkness [133].

Self-healing materials are the class of smart materials that have the ability to restore their original state (either partially or completely) when physically damaged. Depending on the healing mechanism, they can be classified as extrinsic, where a healing agent is released, or intrinsic, where reversible bonds (e.g., non-covalent [134,135] or covalent bonds, including disulfide [136,137] and diselenide bonds [138,139,140,141]) are disassociated and/or reformed under pH and temperature changes or light irradiation. Liu et al. [141] reported visible light-mediated self-healing of protein hydrogels containing diselenide and Schiff base bonds. Healing was evaluated by splitting the material into two pieces, bringing them into contact, and allowing the regeneration of the structural integrity under visible light, excluding any external heat or stress. The hydrogels exhibited healing efficacy of almost 100% after 16 h of irradiation, while the healing in the absence of visible light was only 37%. The impact of selenium content on healing was determined by partially replacing selenium with hexamethylenediamine, which resulted in a decreased rate of self-healing.

## 10. Conclusions

Selenium and sulfur are two elements that share similar electronic configurations and chemical bonding abilities. Because of this, they can often be used interchangeably in chemical reactions. This is particularly true for peptides, where Sec can substitute for Cys and in many cases retain the same biological activity. Despite these similarities, peptides containing Sec show higher reactivity than their Cys-containing analogs. Reactions involving selenopeptides are typically highly selective. The significant acidity of the SeH groups and the nucleophilicity of the selenium atom are responsible for these properties. The relatively low energy required to dissociate the diselenide bond tends to favor radical reactions. In addition, SeH groups are more susceptible to oxidation than thiol groups. These properties have led to many applications of selenopeptides in peptide chemistry. The susceptibility to oxidation allows the selective formation of Se-Se bridges in peptides containing Cys residues, allowing the peptide to be oxidized to its native structure. The use of selenopeptides in NCL is also developing rapidly, as ligations occur more rapidly in systems containing Sec, and products can be selectively deselenated even in the presence of thiol groups in the peptide. The high speed and reversibility of reactions involving selenopeptides make them promising targets in dynamic combinatorial chemistry, where diselenide metathesis in peptides and transamidation of peptides by reversible formation of selenoester bonds may find applications. Continuing research in this area may lead to further discoveries and uses for selenopeptides.

## Data Availability

Not applicable.

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
