# Peer review of "Selenium in Peptide Chemistry"

_molecules, 2023, doi:10.3390/molecules28073198_

Round 1

Reviewer 1 Report

This manuscript describes the overview of selenium in the peptide chemistry. It contains broad range of topics, physical properties and reactivities of selenocysteine, selenocysteine-mediated native chemical ligation and application of selenocysteine in peptide. This review will contribute to the understanding of selenium-based peptide chemistry for researchers in various academic fields. However, I suggest checking the manuscript for any careless mistakes before submission.

Prior to the publication, the following points should be considered.

1. I could not find some studies. I recommend it might be better to cite these important studies. Such as, seleno-azurin (JACS, 2002, 124, 2084-2085), seleno-caenopore-5 (Chem. Sci., 2016, 7, 2005-2010), seleno-conotoxin (JACS, 2010, 132, 3514-3522), seleno-insulin (Chem. Eur. J. 2019, 25, 8513-8521) and so on.

2. Page 2, line 89, or where it first appears:

For "BPTI", it might be better to define the abbreviation by spelling out bovine pancreatic trypsin inhibitor. It looks like BPT1 to me (arabic numerals). If the abbreviation was written as a BPT1, the author must change BPT1 to BPTI correctly in whole manuscript. I found it in the page 2 (line 91 and 93), page 3 (line 97) and page 7 (line 285, 287, 290 and 291).

3. Page 3, line 108, or where it first appears:

Solid-phase peptide synthesis might also be better to define as an abbreviation.

4. Page 3, line 109, or where it first appears:

Native chemical ligation might also be better to define as an abbreviation.

5. Page 3, line 124:

If possible, it may be better to briefly mention the reaction mechanism of NCL as indicated by scheme 2.

6. Page 10, line 415:

"Table 7." should be deleted.

7. In the scheme 2, S-to-N acyl shift is indicated by an equilibrium arrow. However, it is not equilibrium reaction. It must be changed to a single arrow.

Reviewer 2 Report

see attached

Author Response

Please seethe attachment

Round 2

Reviewer 2 Report

While the authors addressed my previous comments, I would encourage them to read through these edits carefully, checking for missing words or added spaces, etc. 

One specific point regarding the new statement: "The access to the SePs is limited, since the translational Sec incorporation is based on the alternative use of the stop codon UGA, resulting in a site-reaction consisting in the termination of transcription [ xi]. Although the alternative approaches have been developed based on the engineering of the Sec-specific tRNA [ xii], this technique is not yet routinely used for SePs expresssion [ xiii]."

Not all UGA codons have Sec inserted, instead there is a SECIS element present which directs Sec to specific UGA codons. While I understand the details of the natural mechanism are not pertinent to the review, the generalization should not overlook this phenomenon. Saying that translation (not transcription) termination is a side reaction is too vague. Moreover, the authors cite a single paper on Sec-specific tRNA engineering from 2015. There have been additional papers since then which have been detailed in doi: 10.1016/j.abb.2022.109421. Also, the statement that suggests using these tRNAs for SeP expression is not routine would be an understatement. There are many examples that have used such technology (10.1021/acs.biochem.9b00973, 10.1073/pnas.2100921118, 10.1002/aur.2655, 10.1016/j.jmb.2021.167199) and given that over 20 requests have gone in for the plasmid system that supports Sec incorporation (Addgene: pSecUAG-Evol2) that will soon increase.

Author Response

While the authors addressed my previous comments, I would encourage them to read through these edits carefully, checking for missing words or added spaces, etc.

Response:

We checked the text and made the suggested corrections

One specific point regarding the new statement: "The access to the SePs is limited since the translational Sec incorporation is based on the alternative use of the stop codon UGA, resulting in a site-reaction consisting in the termination of transcription [ xi]. Although the alternative approaches have been developed based on the engineering of the Sec-specific tRNA [ xii], this technique is not yet routinely used for SePs expresssion [ xiii]."

Not all UGA codons have Sec inserted, instead there is a SECIS element present that directs Sec to specific UGA codons. While I understand the details of the natural mechanism are not pertinent to the review, the generalization should not overlook this phenomenon. Saying that translation (not transcription) termination is a side reaction is too vague. Moreover, the authors cite a single paper on Sec-specific tRNA engineering from 2015. There have been additional papers since then which have been detailed in doi: 10.1016/j.abb.2022.109421. Also, the statement that suggests using these tRNAs for SeP expression is not routine would be an understatement. There are many examples that have used such technology (10.1021/acs.biochem.9b00973, 10.1073/pnas.2100921118, 10.1002/aur.2655, 10.1016/j.jmb.2021.167199) and given that over 20 requests have gone in for the plasmid system that supports Sec incorporation (Addgene: pSecUAG-Evol2) that will soon increase.

Response:

This fragment has been modified to avoid oversimplification of mechanisms related to the expression of proteins containing selenocysteine. However, I tried to avoid overly expanding the text, which is already long. The changes are highlighted in the manuscript (see manuscript).